# Metabolome-Based Discrimination Analysis of Shallot Landraces and Bulb Onion Cultivars Associated with Differences in the Amino Acid and Flavonoid Profiles

**DOI:** 10.3390/molecules25225300

**Published:** 2020-11-13

**Authors:** Mostafa Abdelrahman, Nur Aeni Ariyanti, Yuji Sawada, Fumitada Tsuji, Sho Hirata, Tran Thi Minh Hang, Mami Okamoto, Yutaka Yamada, Hiroshi Tsugawa, Masami Yokota Hirai, Masayoshi Shigyo

**Affiliations:** 1Botany Department, Faculty of Science, Aswan University, Aswan 81528, Egypt; meettoo2000@yahoo.com; 2Department of Biology Education, Faculty of Mathematics and Natural Sciences, Universitas Negeri Yogyakarta, Yogyakarta 55281, Indonesia; nuraeni@uny.ac.id; 3RIKEN Center for Sustainable Resource Science, 1-7-22 Suehiro-cho, Tsurumi-ku, Yokohama, Kanagawa 230-0045, Japan; yuji.sawada@riken.jp (Y.S.); mami.okamoto@riken.jp (M.O.); yutaka.yamada@riken.jp (Y.Y.); hiroshi.tsugawa@riken.jp (H.T.); masami.hirai@riken.jp (M.Y.H.); 4Institute of Food Sciences and Technologies, Ajinomoto Co., Inc., 1-1 Suzukichō, Kawasaki-ku, Kawasaki 210-8681, Kanagawa, Japan; fumitada_tsuji@ajinomoto.com; 5Laboratory of Agroecology, Faculty of Agriculture, Kyushu University, Kasuya, Fukuoka 811-2307, Japan; hirata.sho.481@m.kyushu-u.ac.jp; 6Department of Agronomy, Vietnam National University of Agriculture, Trauqui, Gialam, Hanoi 100000, Vietnam; ttmhang@vnua.edu.vn; 7Laboratory of Vegetable Crop Science, College of Agriculture, Graduate School of Sciences and Technology for Innovation, Yamaguchi University Yamaguchi City, Yamaguchi 753-8515, Japan

**Keywords:** shallots and onions, amino acids, flavonoids, metabolome profiles

## Abstract

Shallot landraces and varieties are considered an important genetic resource for *Allium* breeding due to their high contents of several functional metabolites. Aiming to provide new genetic materials for the development of a novel bulb onion cultivar derived from intraspecific hybrids with useful agronomic traits from shallots, the metabolic profiles in the bulbs of 8 Indonesian shallot landraces and 7 short-day and 3 long-day bulb onion cultivars were established using LC–Q-TOF-MS/MS. Principal component analysis, partial least squares discriminant analysis, and dendrogram clustering analysis showed two major groups; group I contained all shallot landraces and group II contained all bulb onion cultivars, indicating that shallots exhibited a distinct metabolic profile in comparison with bulb onions. Variable importance in the projection and Spearman’s rank correlation indicated that free and conjugated amino acids, flavonoids (especially metabolites having flavonol aglycone), and anthocyanins, as well as organic acids, were among the top metabolite variables that were highly associated with shallot landraces. The absolute quantification of 21 amino acids using conventional HPLC analysis showed high contents in shallots rather than in bulb onions. The present study indicated that shallots reprogrammed their metabolism toward a high accumulation of amino acids and flavonoids as an adaptive mechanism in extremely hot tropical environments.

## 1. Introduction

*Allium* is one of the largest monocot genera in the family Amaryllidaceae, with approximately 850 species that include important vegetable crops, such as onions (*A. cepa*), shallots (*A. cepa* Aggregatum group), leeks (*Allium ampeloprasum*), and garlic (*A. sativum*) [1,2]. *Allium* species are widespread in the northern hemisphere, from the dry tropics to the boreal zone, which has led to the development of an amazing number of *Allium* species with different physiological and morphological traits [3,4]. Due to their medicinal and culinary properties, several plants of the *Allium* genus have been used as vegetables, food seasonings, or folk medicine for long time [5]. S-Allyl-L-cysteine sulfoxide (ACSO) compounds are the most characteristic ingredients in *Allium* species and are principally responsible for the species-specific flavor and biological activities [6]. For example, shallots produce different amounts of ACSOs and polysaccharides as compared with bulb onions [7], and these compounds contribute to the unique hot, pungent taste of shallots. However, other chemical compositions, such as amino acids and ribonucleotides, also play an important role in the taste quality of shallots [8], as an important factor for consumer preference. In addition, the accumulated free amino acids, polyamines, and sugars act as osmoprotectants, which play important roles in the adaptation of plant cells to various adverse environmental conditions by increasing the osmotic pressure in the cytoplasm and stabilizing the membranes and proteins during stress conditions [9]. *Allium* species are also rich sources of different types of flavonoids, especially quercetin and its glycosides, as major pigments found in brown onions, while anthocyanins such as cyanidin and peonidin derivatives are the most abundant pigments found in red onions [10]. Flavonoids play multiple roles in the responses of higher plants to a wide range of environmental constraints by reducing the oxidative damage caused by stress-induced reactive oxygen species (ROS), in addition to their role as a chemical barrier against a wide range of plant pathogens [11,12].

Comparative metabolome analysis of shallots and bulb onions clearly indicated that shallots exhibited a metabolic profile distinct from that of bulb onions, including high total flavonoids, ACSOs, and polysaccharides [7,13]. However, the amino acid profiles of shallots and bulb onions have not been reported. Therefore, in this study, a comprehensive metabolome analysis of 10 Japanese short-day and long-day bulb onion cultivars and 8 Indonesian shallot landraces was initially examined using liquid chromatography–quadrupole time-of-flight-tandem-mass spectrometry (LC–Q-TOF-MS/MS). The 21 amino acid contents in the 10 examined bulb onion cultivars and 12 (9 Indonesian and 3 Vietnamese) shallot landraces were measured using an amino acid analyzer. Finally, expressions of amino acid and flavonoid biosynthesis and regulatory genes in the shallot double haploid (DHA) and bulb onion double haploid (DHC) were compared. Our results provide useful information regarding shallot metabolic traits, which are significant for use in the production of an F_1_ hybrid between shallots and bulb onions to improve the taste and flavor as well as the stress tolerance of bulb onions.

## 2. Results

### 2.1. Comprehensive Metabolite Profiles in Indonesian Shallots and Japanese Long-Day and Short-Day Bulb Onion Cultivars

Bulb samples taken from 8 Indonesian shallot landraces and 10 Japanese long-day and short-day bulb onion cultivars were subjected to non-targeted metabolome analysis using liquid chromatography–quadrupole time-of-flight-tandem-mass spectrometry (LC–Q-TOF-MS/MS). Initially, the mass-to-charge (*m*/*z*) values of the molecular ions of interest that changed between investigated samples were selected and searched against reference databases. Fragmentation spectra for the assigned metabolite signals were collected by tandem mass spectrometry (MS/MS) to verify the identities of the metabolites by comparing the obtained MS/MS spectra and retention times (RTs). Accordingly, a metabolome dataset composed of 251 negative and 389 positive ion mode metabolite signals was developed (Appendix A). In total, 407 metabolite signals were successfully annotated in the 18 examined shallot landraces and bulb onion cultivars on the basis of MS/MS spectra, exact mass values, and RTs, as well as a MS/MS similarity matching against the NIST Mass Spectrometry Data Center and MassBank using an identification score cutoff of 70%, while 233 metabolite signals were unknown (Appendix A).

All of the 640 annotated and unknown metabolite signals were normalized using an internal standard, and the normalized data matrix was subjected to both unsupervised principal component analysis (PCA) and supervised partial least squares discriminant analysis (PLS-DA) (Figure 1A,B). PCA provides an overview of the variation in a dataset; however, predefined treatments/classes are not involved in the PCA algorithm [14]. On the other hand, the PLS-DA algorithm includes predefined treatments/classes, and thus it can maximize the separation among genotypes and maximize the covariance between metabolite variables to enable better understanding of the factors driving genotype separation [14]. Thus, we use both PCA and PLS-DA as a more comprehensive analysis to assess the metabolomic data of shallot landraces and bulb onion cultivars (Figure 1A,B). Both PCA and PLS-DA models of these metabolite signal intensities clearly showed relevant metabolic variations in Indonesian shallot landraces in comparison with short-day and long-day Japanese bulb onion cultivars (Figure 1A,B). In PCA, most of the variations in the metabolite variables in the examined shallots and bulb onions were captured by PC1 (67.7 and 48.8%), especially in the positive ion mode, while the lower proportion of variances (19.2 and 27.7%) were captured by PC2 (Figure 1A,B). With respect to the PLS-DA model, a cross-validation and permutation test showed a high Q2 > 0.93, indicating that the PLS-DA model provides good predications. In addition, the PLS-DA model showed two major clusters, similar to the PCA results, where PC1 captured a large proportion of variances (67.6 and 41.2%) and PC2 displayed a lower proportion of variances (18.6 and 34.7%) (Figure 1A,B). In general, PCA and PLS-DA separated the shallot landraces from short-day and long-day bulb onion cultivars, indicating that the investigated shallot landraces exhibited metabolic profiles distinct from those of bulb onion cultivars (Figure 1A,B). The variable importance in projection (VIP) was also estimated based on the PLS-DA model to identify the key metabolite variables driving the separation between shallots and bulb onions (Figure 1A,B). The VIP scores indicated that amino acids and organic acids such as arginine (Arg, C129), tryptophan (Trp, C161 and C144), proline (Pro, C42), aspartame (C118), butenoic acid (C44), glutamic acid (Glu, C62), pinolenic acid (C215), and methotrexate (C166) were among the top VIPs that might be characteristic metabolites driving the separation between shallots and bulb onions (Figure 1A,B).

Next, Student’s *t*-test and fold change (FC ≥ 2.0 or FC ≤ 0.5; *p*  <  0.05) cutoffs were implemented to identify differentially produced metabolites (DPMs) in the examined shallots and bulb onions using MS-DIAL [15], and volcano plots (Figure 2A; Appendix A). Out of 640 metabolite signal intensities, 77 and 13 increased and decreased metabolites, respectively, were differentially produced in the shallot/onion comparison in positive ion mode, while 97 and 33 increased and decreased metabolites, respectively, were differentially produced in the shallot/onion comparison in negative ion mode (Figure 2A; Appendix A). PatternHunter-based correlation analysis was performed using all of the 220 DPMs identified by positive and negative ion modes as investigated features against shallots/onions as target groups (Figure 2B). Spearman’s rank correlation was selected as the PatternHunter method, and the results showed the top 25 metabolites that exhibited strong correlations with shallots and onions (Figure 2B). Next, genotype–genotype correlation was calculated based on the 220 DPMs (Figure 2C). The results clearly showed high positive correlations (*r* > 0.77 to 0.98) among shallot landraces and high positive correlations (*r* > 0.71 to 0.99) among Japanese bulb onion cultivars, whereas weak correlations were observed among shallot landraces and bulb onion cultivars (Figure 2C). The weak correlations between shallot landraces and bulb onion cultivars provided additional evidence of the distinct metabolic profiles of each genotype. In addition, clustering analysis was performed as a complement to PCA and PLS-DA, and the result showed two major clusters, where all shallot landraces were separated into cluster I and all Japanese bulb onion cultivars were separated into cluster II (Figure 2D), consistent with the PCA and PLS-DA results (Figure 1A,B).

Among the 220 DPMs, several amino acids, organic acids, nucleosides, fatty acids, and flavonoid-related metabolites showed high accumulation in shallots as compared with bulb onions (Figure 3A,B; Appendix A), which could be a characteristic feature of shallots. Specifically, free (Arg, Pro, Trp, asparagine (Asn), aspartic acid (Asp), threonine (Thr), phenylalanine (Phe), Leucine (Leu), glutamine (Gln), glutamic acid (Glu), methionine (Met), serine (Ser), lysine (Lys), and histidine (His)) and conjugated (isoleucine (Ile)-Ile, Ile-Pro, Ile-Asp, pro-valine (Val), and Val-Glu) amino acids and spermidine polyamine were highly accumulated in all shallot cultivars as compared with bulb onions (Figure 3A; Appendix A). Similarly, organic acids such as succinic acids increased in shallots as compared with bulb onions, whereas argininosuccinic and DL-isocitric acids, as well as tyrosine (Tyr) and α-Asp-Lys, were specifically accumulated in bulb onions (Figure 3A; Appendix A). The above results suggested that shallots reprogram their metabolite pools toward a high accumulation of amino acids, which may contribute to shallot adaption to tropical environments [7,13]. In addition, the comparative metabolome profiles of shallots and bulb onions showed a specific accumulation of flavonoids and phenolics in shallot landraces (Figure 3B; Appendix A). For example, quercetin, pelargonin, petunidin, naringin, cosmosiin, *p*-coumaric acid, and catechin were abundant in several shallot cultivars, whereas the methylated chalcone and caffeic acid were accumulated in some bulb onion cultivars (Figure 3B; Appendix A). In addition, some alkaloids and heterocyclic compounds such as trigonelline, julolidine, 1H-indole-4-carboxaldehyde, minocycline, and others were specifically accumulated in shallot landraces as compared with bulb onion cultivars (Figure 3B; Appendix A). The above results indicated that shallots reprogrammed their metabolism to produce more bioactive secondary metabolites, such as flavonoids, which could be a characteristic adaption mechanism for shallot survival under tropical environments.

### 2.2. Differences in Amino Acid Contents between Shallots and Bulb Onions Using an Amino Acid Analyzer

In order to verify the high accumulation of amino acids in shallot cultivars obtained by untargeted metabolome analysis, the absolute contents (mg g^−1^ FW) of 21 amino acids were measured in the bulb tissues of 10 short-day and long-day Japanese onion cultivars, 3 Vietnamese shallot landraces, and 9 Indonesian shallot landraces using an amino acid analyzer (Appendix A). The amino acid contents were classified based on their detection range in both shallots and bulb onions as low, moderate, or high (Appendix A). In the low amino acid group, 7 amino acids, namely β-alanine (β-Ala), Gln, Trp, Met, Tyr, Ile, and Lys, were detected in very low amounts, ranging from 0.0 to 0.05 mg g^−1^ FW in both shallot landraces and bulb onion cultivars (Appendix A). Among these 7 amino acids, Gln showed the lowest content, 0.0 mg g^−1^ FW, in both shallots and bulb onions (Appendix A). In the moderate amino acid group, 7 amino acids, namely His, Thr, Pro, cysteine (Cys), Val, Leu, and Phe, were detected in moderate amounts, ranging from 0.0 to 0.30 mg g^−1^ FW in both shallots and bulb onions (Appendix A). On the other hand, 7 amino acids, namely Asp, Asn, Glu, Ser, Arg, α-Ala, and glycine (Gly), were detected in high quantities, ranging from 0.03 to 1.71 mg g^−1^ FW in both shallot landraces and bulb onion cultivars (Appendix A). The identified amino acids were converted into a KEGG compound ID and mapped onto a KEGG pathway using a KEGG mapper, and the amino acid profiles of shallots and bulb onions measured using the amino acid analyzer were mapped onto the amino acid biosynthesis pathway (Figure 4; Appendix A). In general, most of the shallot landraces exhibited higher amino acid contents as compared with bulb onions (Figure 4; Appendix A). Among the 21 amino acids, Glu, Asn, Ser, His, Thr, α-Ala, Pro, and Val showed higher accumulations in both Indonesian and Vietnamese shallots as compared with bulb onion cultivars, which is consistent with the untargeted metabolome results (Figure 3A; Appendix A). Among higher amino acids, Asn and α-Ala showed the highest contents, especially in shallot landraces, as compared with bulb onion cultivars (Figure 4; Appendix A).

### 2.3. Transcriptome Analysis of Amino Acid and Flavonoid Metabolism in the Shallot Double Haploid (DHA) and Bulb Onion Double Haploid (DHC)

In order to link the metabolome data with the gene level, the transcriptional data of some selected amino acid biosynthesis and transporter genes, as well as the flavonoid biosynthesis and regulatory genes in DHA and DHC, were retrieved from *Allium* TDB (Appendix A). Although this transcriptional data did not include the current examined shallot landraces and bulb onion cultivars, we believe it can give us an overview of the transcriptional level of amino acid and flavonoid-related genes in shallots and bulb onions at the constitutive level (Appendix A). The transcriptome of several amino acid biosynthesis and transporter genes were highly expressed in DHA as compared with DHC, which can explain the high amino acid contents in several shallot landraces (Appendix A). For example, the high level of Pro content in the examined shallot landraces was consistent with the high expression of *delta 1-pyrroline-5-carboxylate synthase 2*, *Delta-1-pyrroline-5-carboxylate dehydrogenase*, *Proline transporter 1*, and *Proline transporter 2* genes involved in proline biosynthesis and transport in DHA versus DHC (Appendix A). Likewise, the high accumulation of Glu and Gln in the examined shallot landraces can be linked with the high expression of several genes involved in Glu metabolism, such as the *Glutamate dehydrogenase 1*, *Glutamate dehydrogenase 3*, and *Glutamate-ammonia ligase* genes, in DHA as compared with DHC (Appendix A). Additionally, several amino acid transporter genes, such as the *Amino acid permease*, *Polyamine transporter*, *Transmembrane amino acid transporter*, *Aromatic and neutral transporter,* and *Branched-chain amino acid aminotransferase* genes, were exclusively upregulated in DHA (Appendix A).

With respect to flavonoids, the transcriptome data showed an increase in the expression level of several genes involved in anthocyanin biosynthesis, such as *Dihydroflavonol 4-reductase-like1, Leucoanthocyanidin dioxygenase*, *Anthocyanidin 5,3-O-glucosyltransferase*, and *TT7/CYP75B1,* in DHA relative to DHC, which was consistent with high levels of pelargonin, petunidin, and cosmosiin in shallots as compared with bulb onions (Appendix A). On the other hand, the highly methylated chalcone in bulb onion cultivars was in line with the increase in the expression of *Chalcone synthase C2*, *Chalcone-flavanone isomerase,* and *Chalcone isomerase* in DHC as compared with DHA (Appendix A).

## 3. Discussion

Our recent studies [7,13,16,17] showed that shallot landraces are crucial genetic resources for *Allium* breeding due to their high contents of variable bioactive metabolites, including oligosaccharides, flavonoids, saponins, and ACSO compounds. For example, the introgression of chromosome 2A from the shallot (AA) to the Japanese bunching onion (*A. fistulosum*, FF) improved *A. fistulosum* (FF2A) disease resistance against Fusarium pathogens, which was attributed to induced–alliospiroside A saponin biosynthesis in the roots of FF2A as compared with the FF genotype [16]. Likewise, comparative metabolome analysis of DHA and DHC showed higher accumulations of amino acids, carbohydrates, phospholipids, and flavonoids in the DHA than in the DHC, reflecting the adaptability of shallots toward abiotic/biotic stress conditions as compared with bulb onions [13]. In addition, the phytochemical compositions of shallot landraces and short-day and long-day bulb onion cultivars demonstrated that Indonesian shallots possessed high amounts of ACSO, especially methiin and isoalliin, as compared with bulb onions, contributing to the high pungency of shallot landraces [7]. The high ACSO in shallots might be a metabolic adaption that enabled shallots to survive under warm and humid tropical environments, acting as an antimicrobial agent against various microorganisms and also as an insect and herbivore repellent [18]. In general, bulb onions and shallots possessed different taste characteristics based on their chemical compositions, which is important for the development of an F_1_ hybrid with high pungency. In addition to ACSO, amino acid compositions represent an important factor in taste development as well as plant survival under harsh environmental conditions [8,19]. For example, transcriptome analysis of two contrasting onion genotypes under drought stress showed an increase in the expression level of the vacuolar amino acid transporter and cationic amino acid transporter genes in a drought-tolerant 1656 onion genotype as compared with the drought-sensitive 1627 [20]. The authors suggested that the increase in total free amino acids serves as potential osmolytes, ROS-scavengers and signaling molecules which collectively enabled drought-tolerant onion genotype to survive under drought stress condition [20].

In this context, the metabolome analysis of 10 short-day and long-day bulb onion cultivars and 8 Indonesian shallot landraces was investigated using LC–Q-TOF-MS, with special focus on amino acid profiles (Appendix A). PCA, PLS-DA, genotype–genotype correlation, and clustering analysis of the normalized metabolite signals revealed distinct metabolic profiles in both shallot landraces and bulb onion cultivars (Figure 1A,B and Figure 2C,D). VIP scores and Spearman’s rank correlation enabled us to identify several metabolite variables that contributed to shallot and bulb onion differentiations (Figure 1A,B and Figure 2B). For example, several amino acids, polyamines, and organic acids were specifically accumulated in shallots as compared with bulb onions, including Arg, Asp, Asn, Pro, Glu, Lys, Thr, Trp, Phe, Ile-Pro, Ile-Ile, Pro-Val, Asp-Pro, Val-Asp, pipecolinic acid, spermidine aspartame, and succinic acids (Figure 3A; Appendix A). The validation of the high amino acid profiles in shallot landraces as compared with those in bulb onion cultivars was further confirmed by an amino acid analyzer, indicating that the high amino acid contents are metabolically characteristic of shallots (Figure 4; Appendix A). The transcriptome data obtained from *Allium* TDB [16] showed a prominent increase in the expression levels of several amino acid biosynthesis and transporter genes in the DHA as compared with the DHC (Appendix A), indicating that shallots reprogrammed the gene-to-metabolite network toward higher amino acid accumulation. The variations in the production of amino acids in shallots as compared with bulb onions could be attributed to genetic and environmental effects [21,22], and thus the examination of further genetic × environmental interactions remains as a future task.

Amino acids and polyamines are well known osmoprotectants that play an important role in stabilizing cellular structures and enzymatic activities against protein denaturation [23]. In addition, these osmoprotectants have been shown to maintain the cell turgor pressure and osmotic potential under a limited water supply and stabilize the redox balance to detoxify the high levels of reactive oxygen species (ROS) under stressful environments [24,25]. The increase in amino acids and polyamines in shallots indicates that shallots reprogram their metabolism to maintain high amino acid content, which could be a prerequisite for shallot growth and survival under warm and hot tropical environments [13]. Thus, the high variation in the levels of amino acids in shallot landraces provides an opportunity to increase the tolerance of bulb onions against harsh conditions by utilizing the shallot landraces with high amino acid production in an *Allium* breeding program. Consistently, monosomic addition lines (MALs) derived from *A. fistulosum* (FF) with an extra chromosome from shallots (AA) showed a higher level of amino acids, especially Gly and Cys, in MALs as compared with FF during the summer season [26]. Additionally, the taste of free amino acids such as Glu and Asp and some selected ribonucleotides is savory, or “umami/delicious” [8], whereas Ser, Gly, Ala, Pro, and sugars contributed to a sweet taste, which collectively make the taste of shallots more unique as compared with bulb onions. In the present study, shallots not only showed higher amounts of amino acids but also a higher level of ribonucleotides, such as isocytosine, uracil, and guanine, as compared with bulb onions (Appendix A), providing additional evidence of the unique chemical composition of shallots, making them more valuable as a new source of food-enhancing compounds. The unique taste of shallots is an important factor in eating quality and consumer preference, as in Southeast Asian markets [7,27].

In addition to ACSOs and amino acids, flavonoids have been detected in different *Allium* species, with cyanidin and peonidin derivatives mainly found in red onions, whereas flavonols, specifically quercetin and its derivatives, are the most abundant ones in the pigment of brown onions [28,29,30]. For example, comparative flavonoid profiles in 31 shallot varieties obtained from Vietnam, Indonesia, Thailand, and the Philippines showed higher phenolic and flavonoid contents in shallot varieties originating from the low-latitude regions than shallots from high latitudes, and quercetin, quercetin-4′-*O*-monoglucoside, and quercetin-3,4′-*O*-diglucoside were the most abundant flavonoid compounds detected in the investigated shallots [30]. Similarly, the comparative total flavonoid contents in shallot landraces and bulb onion cultivars indicated that Indonesian shallots exhibited a higher amount of total flavonoids as compared with bulb onion cultivars [7]. The high flavonoid and phenolic contents in shallots were also positively correlated with high antioxidant capacity as compared with bulb onions [28]. In the present study, several flavonols and anthocyanins were accumulated in shallot landraces, as compared with bulb onion cultivars, and the increase in flavonoid contents in shallots was highly coordinated with the high expression of several flavonol and anthocyanin biosynthesis and regulatory genes in the DHA compared with the DHC (Figure 3B and Appendix A). Flavonoids are potent antioxidants that play prominent roles in protecting plant cells against ROS, generated from photosynthesis and the respiration process under normal and stressed conditions, that cause cellular damage to membranes, proteins, and nucleic acids, as they contain unpaired electrons [31]. In addition, flavonoids located mainly in the outer skin of the bulb tissue act as a chemical barrier against pathogens [10]. Therefore, shallot landraces with high flavonol and anthocyanin contents can be used for *Allium* breeding to improve the flavonoid contents in bulb onions and increase their tolerance against abiotic and biotic stresses. Consistently, *A. fistulosum* with an extra chromosome 5A (FF5A) from shallots exhibited high flavonol and anthocyanin contents, and this was attributed to the upregulation of several flavonoid biosynthesis and regulatory genes in FF5A as compared with the FF genotype [17].

In conclusion, the investigated shallot landraces showed higher amino acid and flavonoid contents than short-day and long-day bulb onions when using both the untargeted metabolome and amino acid analyzer detection methods. The increase in amino acids and flavonoids might be a metabolic adaption mechanism in shallots for survival in tropical environments. The shallot landraces investigated in this study can be useful genetic materials for bulb onion breeding to produce an F_1_ hybrid with high amino acid and flavonoid contents as important agronomic traits.

## 4. Materials and Methods

### 4.1. Plant Materials

In this study, 20 mg dry weight of bulb samples derived from 10 Japanese short-day and long-day bulb onions and 8 Indonesian shallot landraces was extracted and analyzed using the previously described methods. Metabolome data were obtained via 54 runs, from which signal intensity data were prepared. The details of each analytical method are described in the Appendix A. Ten Japanese bulb onion cultivars, including 7 short-day and 3 long-day bulb onion cultivars, were used in this study. The short-day bulb onions were cultivated in Kagawa Prefecture (N 34°, E 134°), and the long-day bulb onions were grown in Hokkaido (N 44°, E 142°). In addition, 8 shallot landraces obtained from Indonesia (S 6°, E 106°) and 3 shallot landraces obtained from Vietnam (N 21°, E 105°) were collected from farmers and local markets. All 8 shallot landraces obtained from Indonesia were collected directly from the farmers at Java island in Indonesia, which characterized by Regosol soil type. The cultivation time start in May and harvested in July (55–60 days after transplanting) after all plants reached full maturity stage, which is known by yellowing of the leaves. Manual irrigation and inorganic fertilizers were applied during the cultivation period every week. Vietnamese shallots were cultivated in pots filled with sandy soil at the Greenhouse of Yamaguchi University in May and harvested after all the plants reached full maturity. Manual irrigation and inorganic fertilizers were applied during the cultivation period.

Samples were vacuum dried using vacuum dryer without skin. The skin was peeled before the drying. The freeze-dried samples then grounded using small copper/blender and the dry powders were stored at −21 °C before analyses.

### 4.2. Untargeted Metabolome Analysis

In this study, 10 Japanese short-day and long-day bulb onions and 8 Indonesian shallot landraces were used for LC–Q-TOF-MS/MS analysis. Metabolome data were obtained via 54 runs derived from three biological replicates from each plant, from which signal intensity data were prepared. The lyophilized powdered bulb onion samples were weighed at 20 mg, and the powdered samples were extracted by a bead shocker (Shake Master Neo, BioMedicalScience) at 1000 rpm for 2 min with 3 mm zirconia beads and 1 mL of solvent (methanol:water = 20:80% with 0.1% formic acid and internal standards: Positively charged ion, 8.4 nM of lidocaine; negatively charged ion, 210 nM of camphor sulfonic acid) in a 1.5 mL tube. The extracted solution was centrifuged by 104 × *g* at 4 °C. The centrifuged 1.5 mL tube was put into an automated liquid handling system (Microlab Star Plus, Hamilton) for dispensing, plate transfer, drying solvent, dissolving, and filtration. The 200 μL of extracted solvent was dried and dissolved by 100 μL of water, then filtrated with a 384-well formatted filter (Multiscreen HTS 384-Well HV, Merk). The 1.5 μL of filtrated solution (40 mg DW sample/mL) was injected for LC–Q-TOF-MS/MS analysis. The setting parameters for LC–Q-TOF-MS/MS analysis are summarized in Appendix A.

The raw data (m/z and intensity values) of LC–Q-TOF-MS/MS analysis were obtained using a Compass Hystar (Bruker), and the m/z values were calibrated using an LC/MS calibrant mix (G1969–85000, Agilent Technologies). The raw data were converted to an Analysis Base File (ABF) using an AbfConverter (Reifycs, http://www.reifycs.com/AbfConverter/). Using the ABF files, MS and MS/MS data were analyzed via MS-DIAL [15,32,33]. All data set can be download from DROP Met (http://prime.psc.riken.jp/menta.cgi/prime/drop_index). The data matrix of MS-DIAL was analyzed using Metaboanalyst v. 4 for PCA, PLS-DA, VIP, and the identification of DPMs. Volcano plots, Pearson’s correlation, and heatmap hierarchal clustering were generated by R. v. 3.5.1 (https://www.r-project.org).

### 4.3. Extraction and Determination of Amino Acid Using Amino Acid Analyzer

The method of extraction used was described by Vu et al. [30]. A 100 µL sample extract and 100 µL of 21 amino acid standards were vacuum-dried separately using a Spin Dryer Lite VC-36R (Taitec Co., Ltd., Saitama, Japan). For derivatization, 20 µL of freshly prepared methanol:water: triethylamine:phenyl isothiocyanate (7:1:1:1, v:v:v:v) was added and mixed with the vacuum-dried extract. The mixture was incubated for 20 min at room temperature and dried under reduced pressure. The derivative mixture was dissolved in 100 µL of 5 mM sodium phosphate, pH 7.6, containing 5% acetonitrile. Fifty microliters of the sample and standards was injected into an amino acid analyzer as described by [26]. The HPLC analysis was run using the following solvent system. Solvent A: 19 g of sodium acetate trihydrate and 2 mL of TEA were dissolved in 1 L of high-purity water. By adding glacial acetic acid, the solution was adjusted to pH 6. Solvent B: 60% acetonitrile and 40% high-purity water (*v*/*v*) were mixed. The transmission of the gradient elution and the flow rate were obtained as described by [26]. All of the amino acid standards were obtained from Sigma-Aldrich, Inc. (Tokyo, Japan) except for Cys, which was purchased from Ajinomoto Co., Inc. (Tokyo, Japan), and the His, The, Tyr, Met, and Phe standards, which were purchased from Wako Pure Chemical Industries, Ltd. (Osaka, Japan). The absolute contents of amino acids were expressed as mg g^−1^ FW using amino acid standard curves. This data was obtained during doctoral study [34].

## Figures and Tables

**Figure 1 molecules-25-05300-f001:**
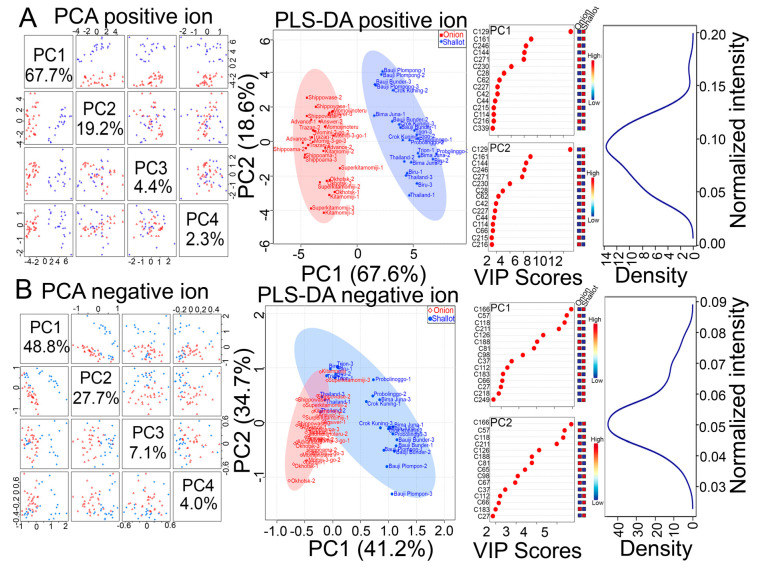
Principal component analysis (PCA) and partial least squares discriminant analysis (PLS-DA) of the assigned metabolite signal intensities obtained from the bulb samples of 10 Japanese short-day and long-day bulb onion cultivars and 8 Indonesian shallot landraces. (**A**,**B**) Score plots of variable importance in projection (VIP) and density distribution of metabolite signal intensities detected by using positive (**A**) and negative (**B**) ion modes of LC–Q-TOF-MS.

**Figure 2 molecules-25-05300-f002:**
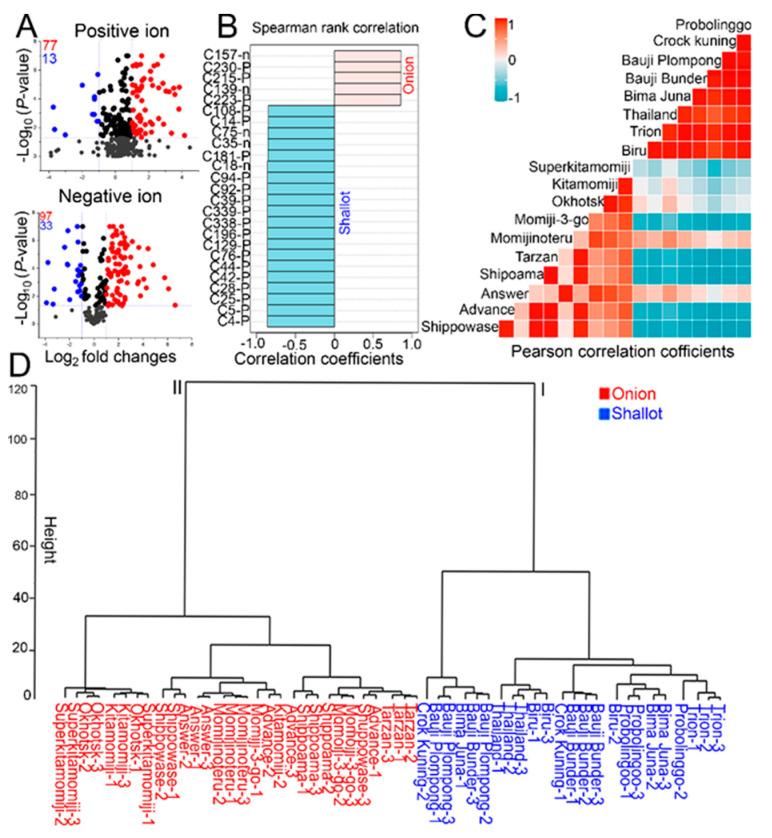
Metabolite profiles and correlation analyses of 10 Japanese bulb onion cultivars and 8 Indonesian landraces. (**A**) Volcano plots of the differentially produced metabolites (DPMs) in the examined bulb onion and shallot samples by positive and negative ion modes of LC–Q-TOF-MS. Red and blue numbers indicate increased (FC ≥ 2.0; *p*  <  0.05) and decreased (FC ≤ 0.5; *p*  <  0.05) metabolites in the shallot/onion comparison, respectively. (**B**) Top 25 metabolites that exhibited a strong positive correlation with shallots and bulb onions based on Spearman’s rank correlation. (**C**) Genotype–genotype correlations of the examined bulb onions and shallot cultivars using Pearson’s correlation coefficients. (**D**) Clustering analysis of the investigated bulb onion cultivars and shallot landraces.

**Figure 3 molecules-25-05300-f003:**
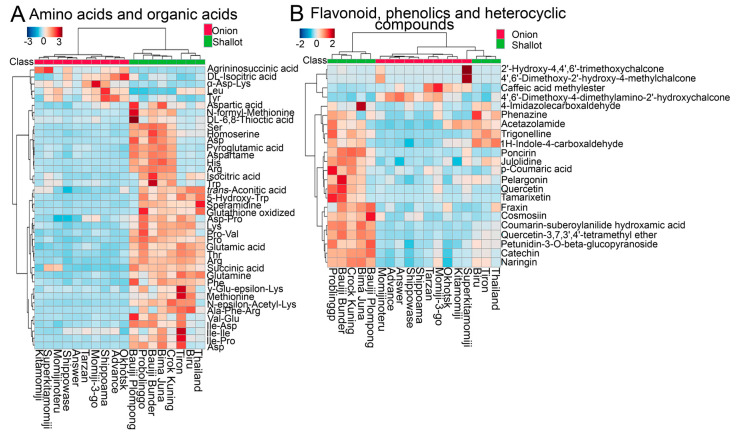
Heatmap hierarchical clustering of (**A**) amino acids and organic acids and (**B**) flavonoids, phenolics, and heterocyclic compounds identified in short-day and long-day Japanese bulb onion cultivars and Indonesian shallot landraces. The heatmap expression was generated using normalized metabolite signal intensities.

**Figure 4 molecules-25-05300-f004:**
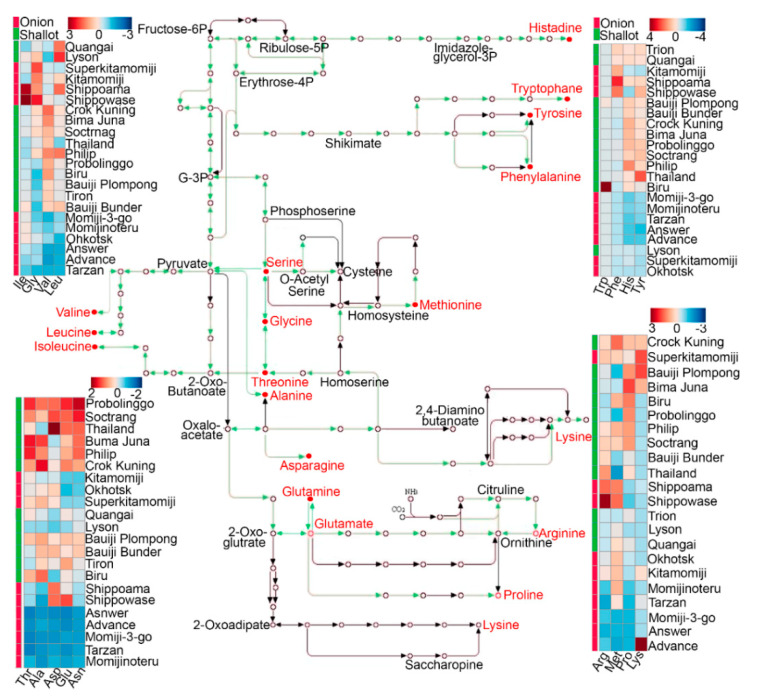
Amino acid profiles of bulb onion cultivars and shallot landraces measured using an amino acid analyzer. The amino acid profiles in the investigated genotypes were mapped onto the amino acid biosynthesis pathway derived from the Kyoto Encyclopedia of Genes and Genomes (KEGG) database.

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
