# Peer review of "Metabolome-Based Discrimination Analysis of Shallot Landraces and Bulb Onion Cultivars Associated with Differences in the Amino Acid and Flavonoid Profiles"

_molecules, 2020, doi:10.3390/molecules25225300_

Round 1

Reviewer 1 Report

This manuscript describes the metabolite differences between shallot landraces and bulb onions, particularly in amino acids and flavonoids.

It is very interesting study and manuscript is well organized.

My major concern is

Plant metabolites such as amino acids, organic acids and flavonoids, their contents are heavily influenced by environmental/geographical condition and also by development stages. Even in the same developmental stage, collection time can also affect the amounts of primary/secondary metabolites. Author examined very limited number of samples without many biological replications. It is hard to conclude whether the differences are due to the different genotype or something else. There are so many other factors that could influence on the metabolites.

All important information on the samples are missing.  

  • Scientific name of the species
  • where/when sample was harvested.
  • how to dry, how to prepare with/without skins…etc.

Those questions have to be answered before publication.

Author Response

#REV.1

This manuscript describes the metabolite differences between shallot landraces and bulb onions, particularly in amino acids and flavonoids.

It is very interesting study and manuscript is well organized.

Response: We highly appreciate the Reviewer for his/her positive evaluation of our manuscript, and providing constructive comments that have helped us greatly improve the quality of the manuscript. We have taken all efforts to revise the manuscript taking into account all the comments and suggestions raised by the Reviewer.

My major concern is

Point1 Plant metabolites such as amino acids, organic acids and flavonoids, their contents are heavily influenced by environmental/geographical condition and also by development stages. Even in the same developmental stage, collection time can also affect the amounts of primary/secondary metabolites. Author examined very limited number of samples without many biological replications. It is hard to conclude whether the differences are due to the different genotype or something else. There are so many other factors that could influence on the metabolites.

Response: We highly appreciate the Reviewer for these critical comments. We totally agree with reviewer concern regarding the environmental effects, which might induce the accumulation of certain metabolites as adaptation mechanism. However, the increase in amino acids in shallots compared with bulb onions has been reported in our previous study (Abdelrahman et al., 2015, Molecular breeding) and both species were grown under same conditions and harvested at same developmental stage, indicating that shallots are able to reprogramed their metabolism towards higher amino acids as an adaption mechanisms due to long-term cultivation in tropical environments. In the present study, shallots were collected from Indonesia, whereas bulb onions and Vietnamese shallots were grown in Japan under greenhouse conditions. However, plants were cultivated and harvested at almost same developmental stage (full maturity). In addition, manual irrigation and inorganic fertilizers were applied for both shallots and bulb onions.

Following the reviewer’s advice all the information related to growth condition of shallot samples and Japanese bulb onion cultivars have been revised in the material and method section as below:

All 8 shallot landraces obtained from Indonesia were collected directly from the farmers at Java island in Indonesia, which characterized by Regosol soil type. The cultivation time start in May and harvested in July (55-60 days after transplanting) after all plants reached full maturity stage, which is known by yellowing of the leaves. Manual irrigation and inorganic fertilizers were applied during the cultivation period every week. Vietnamese shallots were cultivated in pots filled with sandy soil at the Greenhouse of Yamaguchi University in May and harvested after all the plants reached full maturity. Manual irrigation and inorganic fertilizers were applied during cultivation period. (L376-383).

We also would like to confirm that all samples were collected in three biological replicates as indicated in the “Material and Method” section.

Point 2 All important information on the samples are missing.

Scientific name of the species

Response: The scientific name of shallots and bulb onions are indicated first mentioned.

Point 3 where/when sample was harvested.

Response: The harvesting time and sample collections are indicated in the revised manuscript in the “Material and method” section. (L376-383)

Point 4 how to dry, how to prepare with/without skins…etc.

Response: We highly appreciated the Reviewer for this critical comments. Following the Reviewer’s advice, sample preparation has been revised in the “Material and Method” section as below:

Samples were vacuum dried using vacuum dryer without skin. The skin was peeled before the drying. The freeze-dried samples then grounded using small copper/blender and the dry powders were stored at -21°C before analyses. (L384-386)

Reviewer 2 Report

This study presents the comparative metabolomic analysis of oignons and indonesian shallots cultivars. The study and manuscript were very well prepared. 

Couple important comments:

  • The description of the annotation method for metabolites is not sufficiently detailed in the manuscript. Altough the Supplementary tables are providing a variety of informations to understand the method, some aspect are still unclear. For example, some metabolites appeared to have been annotated based on MS1 only. When annotated on MS1, was a specific compound/RT database used ? Also the "isotopic parent ID" is always -1 in positive ion mode of Table S2 ? Shouldn't that information be useful to validate the putative annotation ? Other annotation were conducted based on MS1 and MS2. For MS2 annotation, was it against a spectral library ? Or a in silico annotation. I would suggest to provide more information on these aspects in the method about the annotation method that would help to better understand the level of confidence of the annotation/identification. 
  • My biggest reserve regards the lack of mass spectrometry data sharing that reduce the relevance of the research for the community. The LC-MS/MS data themselves should be made available on MetaboLights or MassIVE or other data repository. If the authors can't share the data because of commercial agreement, a statement of conflict of interest must be added.

Some minor observations:

L86 - English - “The assigned metabolite signals were then subjected to a tandem mass spectrometry to verify …” -> maybe "tandem mass spectrometry experiment" or  “Fragmentation spectra for the assigned metabolite signals were collected by tandem mass spectrometry to verify …”

L92 - "a blast search against the NIST Mass Spectrometry 92 Data Center"
  L317 - "The increase in amino acids and  in shallots indicate” -> “indicates”

Author Response

#REV.2

This study presents the comparative metabolomic analysis of oignons and indonesian shallots cultivars. The study and manuscript were very well prepared.

Response: We highly appreciate the Reviewer for his/her positive evaluation of our manuscript, and providing constructive comments that have helped us greatly improve the quality of the manuscript. We have taken all efforts to revise the manuscript taking into account all the comments and suggestions raised by the Reviewer.

Couple important comments:

Point 1 The description of the annotation method for metabolites is not sufficiently detailed in the manuscript. Altough the Supplementary tables are providing a variety of informations to understand the method, some aspect are still unclear. For example, some metabolites appeared to have been annotated based on MS1 only. When annotated on MS1, was a specific compound/RT database used ?

Also the "isotopic parent ID" is always -1 in positive ion mode of Table S2 ? Shouldn't that information be useful to validate the putative annotation ? Other annotation were conducted based on MS1 and MS2. For MS2 annotation, was it against a spectral library ? Or a in silico annotation. I would suggest to provide more information on these aspects in the method about the annotation method that would help to better understand the level of confidence of the annotation/identification.

Response: We thank the Reviewer for this constructive comment. In this manuscript, we carried out LC-QTOF-MS analysis of MS1 and MS2 (data dependent MS/MS analysis).We only used MS1 and MS2 information for DB search. The "parent ID" function is used for stable isotopic ion analysis of fully labeled samples. In this study, we only used intact sample, thus we deleted the column "parent ID" in supplemental Table. The annotation of this manuscript corresponded to Level 2 of metabolomics standards initiative (MSI). We used public MS/MS database (http://prime.psc.riken.jp/compms/msdial/main.html#MSP).

Point 2 My biggest reserve regards the lack of mass spectrometry data sharing that reduce the relevance of the research for the community. The LC-MS/MS data themselves should be made available on MetaboLights or MassIVE or other data repository. If the authors can't share the data because of commercial agreement, a statement of conflict of interest must be added.

Response: Thank you very much for your constructive comment. We would like to confirm that all data set can be download from DROP Met (http://prime.psc.riken.jp/menta.cgi/prime/drop_index). (L406-407)

Some minor observations:

Point 3 L86 - English - “The assigned metabolite signals were then subjected to a tandem mass spectrometry to verify …” -> maybe "tandem mass spectrometry experiment" or  “Fragmentation spectra for the assigned metabolite signals were collected by tandem mass spectrometry to verify …”

Response: Thank you very much for this constructive comment. Following the Reviewer’s advice, we changed the sentence in the revised manuscript as below:                                                                                                         “Fragmentation spectra for the assigned metabolite signals were collected by tandem mass spectrometry (MS/MS) to verify the identities of the metabolites…” (L389-390).

Point 4 L317 - "The increase in amino acids and  in shallots indicate” -> “indicates”

Response: We are very sorry for this error, which has been checked and revised as below:

The increase in amino acids and polyamines in shallots indicates…” (L319).